# Ethnobotanical Survey of Plants Used for Treating Cough Associated with Respiratory Conditions in Ede South Local Government Area of Osun State, Nigeria

**DOI:** 10.3390/plants9050647

**Published:** 2020-05-20

**Authors:** Ibraheem Oduola Lawal, Ikeolu Idowu Olufade, Basirat Olabisi Rafiu, Adeyemi O. Aremu

**Affiliations:** 1Biomedicinal Research Centre, Forestry Research Institute of Nigeria, Private Bag 5054 Forest hill, Ibadan 200272, Oyo State, Nigeria; rafiu.bo@frin.gov.ng; 2Federal Polytechnic Ede, PMB, 231, Ede 232101, Osun State, Nigeria; idowi.i@fedpoede.edu.ng; 3Indigenous Knowledge Systems Centre, Faculty of Natural and Agricultural Sciences, North-West University, Private Bag X2046, Mmabatho 2790, South Africa

**Keywords:** African traditional medicine, decoction, Fabaceae, fidelity level, herbs, trees, indigenous knowledge systems

## Abstract

In many developing countries, community members depend on their local flora for treating diverse ailments including those affecting the respiratory system. This is often attributed to the high cost and limited access to health care facilities. This present study focused on the documentation of plant species used against cough associated with the respiratory diseases in Ede South Local Government Area of Osun State. The survey was conducted using semi-structured interviews among 100 participants. Information obtained was analyzed using different ethno-botanical indices including relative frequency of citation (RFC) and fidelity level (FL). A total of 87 plant species from 39 families, which was mostly represented by Fabaceae, were reported in the study area. *Crinum jagus* was the most popular plant used against cough and approximately 32% of the plants have been reported as cough remedies for the first time. However, some of the documented plants have been reported for the treatment of cough and related respiratory diseases in several countries. In terms of the life-form, trees constituted the highest proportion of the medicinal plants (37%), while leaves (36%) were the predominant plant part prescribed for cough. Decoction was the main method of preparing the plants, which were all administered orally. Approximately 63% of the plants were exclusively sourced from the wild. The current study revealed the richness and widespread use of plant species for managing cough associated with respiratory diseases in the study area. The generated inventory contributes to the expanding database of valuable plant resources with medicinal potential in Nigeria and Africa.

## 1. Introduction

Respiratory diseases entail conditions such as chronic obstructive pulmonary disease (COPD), asthma, occupational lung diseases, and pulmonary hypertension [1]. Globally, these diseases affect millions of people across diverse ages and account for significant levels of disability and mortality especially in children less than 5 years [2,3,4,5]. The occurrence and severity of respiratory diseases remain high in both developed and developing countries. For instance, it remain one of the four major contributors to mortality and morbidity, resulting in high health cost and loss in productivity in developed countries [4]. Likewise, the disease burden associated with respiratory infections continues to increase in developing countries. Respiratory diseases often arise as a result of air pollution, lifestyle and microbial infection while the risk factors include tobacco smoke, air pollution, occupational chemicals, and dusts as well as frequent lower respiratory infections during childhood [4].

As a common symptom associated with respiratory diseases, cough may be categorized as either acute or chronic in nature [6]. Acute cough represents the largest single cause of consultation in primary care, whereas chronic cough is one of the commonest presentations in respiratory diseases [7,8]. As reported by Chang et al. [9] and Desalu et al. [8], the frequency and severity of cough often varies across different age groups and gender. It is considered to be mild when it acts as a significant preventive or defensive way of securing nasal discharge and other noxious substances from the respiratory tract [9,10]. Given that cough occurs in other non-respiratory conditions including nasal disease, the global market in cough treatments is continuously increasing and runs into several billion dollars [6]. On the basis of the widespread of cough among different populations globally [8,11,12,13,14], the need for improvement in the treatment and management remains pertinent [2,15].

Despite the advances in pharmaceuticals and generally drug development, the importance of natural resources especially plants for healthcare and general well-being cannot be overemphasized [16,17,18,19]. For instance, Alamgeer et al. [20] alluded to the widespread use of plants for treating respiratory diseases in Pakistan. The authors observed that the high incidence of cough is treated with about 56% of the 384 plant species used for various respiratory diseases. In sub-Saharan Africa, the high reliance and widespread utilisation of plants against diverse ailments is common among the different ethnic groups [21,22]. Particularly, the use of plants for respiratory and related diseases have been documented in countries such as Gabon [23], Democratic Republic of the Congo [24,25], Mauritius [26], South Africa [27,28], and Nigeria [29,30,31]. The rich flora and associated indigenous knowledge among different communities have been recognised by researchers in Nigeria [32,33]. Furthermore, the use of medicinal plants is well-recognized among the Yorubas of Southwestern Nigeria. For example, ethnobotanical surveys for plant species used against different disease conditions including malaria [34,35], tuberculosis [36], fever, rheumatism [35], and asthma [31] have been documented. Despite the increasing interest and effort in documenting plant resources with therapeutic value, several research gaps still exists in Nigeria [37]. For instance, the neglect of some communities and diseases/conditions has been observed. Extensive documentation remain important to mitigate the loss of valuable indigenous knowledge associated with plant resources in local communities. Furthermore, the socio-cultural uniqueness and inherent dynamics associated with the understanding of local and indigenous use of medicines among local communities cannot be overlooked [31,38]. Hence, the current study explored the plant species utilised for cough associated with respiratory diseases among the community members in Ede South Local Government Area (Osun State, Nigeria).

## 2. Materials and Methods

### 2.1. Study Area

This survey was conducted at Ede South Local Government Area of Osun State, which is located in the south-western region of Nigeria (Figure 1). The headquarter of the study area is located at Oke Iresi and 242 km^2^ on the latitude 7°42′ N and longitude 4°27′ E. Ede South Local Government Area comprises of 10 wards namely; Babanla Agate (Obada side), Kuye (Orita Oloki), Jagunjagun (Akala side), Alajue 1 (Agip to sawmill and Cottage junction axis), Alajue 11 (village), Olodan (village), Baba Sanya (CAN Tower), Oloki Akoda (along Ibadan express way), Sekona (town), Loogun (permanent site, Federal Polytechnic, Ede and Adeleke University).

The population of the study area is estimated at 76,035 comprising of 39,385 males and 36,654 females [39]. The area is predominantly occupied by the Yoruba-speaking ethnic group. The study area as part of Ede region has drainage systems that ranged from open water bodies (dams, reservoirs and lakes) to rivers, streams, springs, wells, run-off waters and swamp/wetlands [40]. This inevitably influences the micro-climate and vegetation in the study area, which is considered as rain forest [41]. The people in the study area are mostly farmers owing to the availability of green vegetation [42]. Although 70% of the population are living in the villages, the remaining 30% reside in town, engaging in commercial services.

### 2.2. Field Interview Methods

Semi-structured questionnaires were used to obtain the ethnobotanical information from the participants from April to August, 2017. The questionnaire was designed to capture the following information of plants used for the treatment of cough associated with respiratory diseases; local name, used plant part(s) and preparation method. Bio-data of the participants including their age, gender, residence, occupation, and educational background were also recorded. The participants were purposively selected comprising herb-sellers, traditional medical practitioners (herbalist), farmers, and hunters in the area. They were informed of the concept of the study in order to seek their consent and willingness to participate in the survey. The participants were individually questioned on their knowledge of using plant species to treat cough associated with respiratory diseases. Ten native field assistants were engaged to administer and interpret the questions to the participants in their local language (Yoruba) in order to facilitate efficient communication.

Sixty-five men and 35 women (to make-up the 100 participants) were interviewed on their knowledge about cough and its management in the study area. The majority of the participants were males (65%) and mostly herbalists (35%) in terms of their occupation (Table 1).

### 2.3. Plant Collection and Identification

The plant species were initially identified using their local ‘Yoruba’ names and later matched with their respective scientific classifications through consultation of relevant literature [32], while the current taxonomic classification was validated using “The Plant List” [43]. The voucher specimens were prepared, identified and authenticated by an expert prior to depositing at the Forest Herbarium, Ibadan (FHI), Oyo State, Nigeria.

### 2.4. Ethnobotanical Data Analysis

The information (for e.g., plant name and plant part used to treat cough associated with respiratory diseases) obtained through the ethnobotanical interviews were analyzed. In order to establish the importance of the documented plants used in treating cough associated with respiratory diseases, we analysed the data using the following ethnobotanical indices.

Based on the methods described by Tardío and Pardo-de-Santayana [44], we evaluated the Relative frequency of citation (RFC) for plant species mentioned in the study area.
RCF = FC/N(1)
where FC = frequency of citation/mention, N = Number of participants in the survey.

In an attempt to determine the preference of a particular plant species for the treatment of cough in the study area, the fidelity level (FL) was calculated as described by Friedman et al. [45]. High FL indicates high usage of a plant for cough while low FL denotes a low frequency for this condition.
FL = (Ip/Iu) × 100(2)
where lp = number of participants that claimed a use of certain plant species to treat a cough, Iu = total number of participants in the survey.

### 2.5. Ethical Consideration

The study area is within the jurisdiction to the research study zones of Forestry Research Institute of Nigeria [46], a national research platform. The study was approved by the research coordinating unit of FRIN. All the participants provided verbal consent prior to data collection.

## 3. Results and Discussion

### 3.1. Inventory of Plant Species Used against Cough Associated with Respiratory Diseases

In the current study, 87 locally used medicinal plant species from 39 families were recorded for the treatment and management of cough associated with respiratory diseases in the study area (Table 2 and Appendix A). The relative high number of plant species is an indication of the rich flora available for managing cough associated with respiratory conditions among the community members. In view of the diverse etiology of cough, it remains one of the commonest symptoms associated with clinic/hospital visits in many developed countries [6]. Although the epidemiological studies on cough are currently scanty in sub-Saharan Africa [8], it remains one of the conditions that are treated with African medicinal plants [24,25,29,35,47,48]. For instance, cough ranked (5th) among the top 10 diseases commonly treated with medicinal plants among the 930 households sampled in Akwa Ibom State of Nigeria [49]. A study conducted among the rural people in northern Maputaland (South Africa) by York et al. [27], specifically identified cough as a common symptom that is often treated with medicinal plants used for respiratory diseases. Similarly in South Africa, an estimated 37% (2nd to tuberculosis which was the highest at 40%) of the 306 plants used for managing respiratory diseases by traditional healers were targeted at mitigating cough [50]. As revealed by Bekalo et al. [51], cough is a condition that is treated with different medicinal plants by the local people in the lowlands of Konta Special Woreda in Ethiopia. Furthermore, cough was the most common respiratory-tract infection managed using plant species in Mauritius [26]. Beyond Africa, similar utilisation of medicinal plants for counteracting cough have been documented in countries such as India [52], Spain [53], Greece [54], and Yemen [55]. As demonstrated in these aforementioned examples, there is continuous and widespread utilisation of medicinal plants for mitigating cough as a condition and as part of associated symptoms in many diseases especially those affecting the respiratory system.

The family Fabaceae was the most represented for the 87 plant species documented as remedy against cough associated with respiratory diseases (Figure 2). The number of plant species from the family Fabaceae was two-fold higher when compared to the next family with a high number of plants. Five (5) plant families including Asteraceae, Curcurbitaceae, Euphorbiaceae, Malvaceae, and Poaceace had four to five plant species while an estimated 49% of the 39 families had one representative member each. The dominance of the Fabaceae as the most common family to have been widely observed in several ethnobotanical surveys in Nigeria [34,47,56] and other countries in Africa countries [28,51,57,58]. Evidence to support the dominance of Fabaceae as a plant family with medicinal value was recently hypothesized by Van Wyk [59]. This assertion was based on the analysis of 4576 vascular plant species from 1518 genera that are used in Traditional African Medicine in sub-Saharan Africa. In addition, the Fabaceae with other families such as Apocynaceae, Burseraceae and Rubiaceae ranked as the most commonly-traded African medicinal plant species, an indication of commercial value. The popularity and high preference of the members of the family Fabaceae in African Traditional Medicine may be attributed to their availability and abundance as well as adaptability to different environments [56].

### 3.2. Diversity and Ethnobotanical Indices of Plant Species Used against Cough Associated with Respiratory Diseases

Despite the relative high number of plants utilised for cough purpose, their ethnobotanical indices including FC (1–12), RFC (0.01–0.12) and FL (1–40%) were generally low (Table 2). Furthermore, approximately 89% of the plant species had relatively lower indices (for e.g., FC ≤ 3; RFC ≤ 0.03, FL ≤ 3%) when compared to other plants used for cough. In the study area, *Crinum jagus* was the most popular plant used for cough among the participants. Member of the genus *Crinum* have been extensively utilised among diverse diseases in folk medicine globally and an increasing interest from pharmaceutical sector based on the therapeutic potential [60]. Locally known as ‘Ogede odo’, *Crinum jagus* has long been regarded as a potent remedy for relieving asthma and related cough among the Yoruba of south-western part of Nigeria [30,31]. Member of the genus *Crinum* are known to often be used for diverse ailments including respiratory diseases in Democratic Republic of the Congo [24,25], Ethiopia [61], Nigeria [47], and Cameroon [62]. The alkaloidal constituents, which are often characteristically of the family Amaryllidaceae including the genus *Crinum*, are known to significantly contribute to their diverse medicinal attributes [60].

In addition to *Crinum jagus*, *Anacardium occidentale* and *Khaya grandifoliola* (FC = 11; RFC = 0.11; FL = 11%) as well as seven plants (*Abrus precatorius*, *Aframomum melegueta*, *Bridelia ferruginea*, *Citrus aurantifolia*, *Garcinia kola*, *Jatropha curcas*, *Jatropha multifida*, *Cymbopogon citratus*, *Ageratum conyzoides* and *Alstonia boonei* with FC = 10; RFC = 0.1; FL = 10%) were the 10 most common plants used as cough remedy in the study area (Table 2). According to Sonibare and Gbile [31], herbalists and traditional medical practitioners recognised the majority of these aforementioned plants as remedy against asthma and other respiratory conditions in Nigeria.

From the current findings, an estimated 46% and 43% of the plant species have been reported for respiratory-related conditions in Nigeria and other countries, respectively (Table 2). Some of these plants are known to be used for treatments of cough and associated respiratory diseases/conditions (for e.g., asthma, expectorant, tuberculosis and bronchitis) in at least 15 countries globally. For instance, the use of *Psidium guajava* for the treatment of cough has been documented in Pakistan [63], Uganda [64,65], South Africa [27], and Zimbabwe [66]. On the other hand, *Garcina kola* has been extensively documented as a cough remedy in Nigeria [29,36,49] but no record has been found in other parts of the world. Furthermore, reports of the use of approximately 32% of the plants such as *Corchorus olitorius*, *Hybanthus enneaspermus* and *Theobroma cacao* as cough remedy in folk medicine were not found (Table 2). These findings clearly establish the existence of some degree of similarities and uniqueness with regards to the use of plants for treating and managing common ailments among different ethnic groups globally.

### 3.3. Life-Forms and Plant Parts Used against Cough Associated with Respiratory Diseases

In the study area, trees had the highest proportion (37%) while climbers were the lowest life-form for the plant species used for treating cough associated with respiratory problems (Figure 3). The dominance of woody plants (trees and shrubs) was evidence as they accounted for approximately 61% of the plants documented. The popularity of woody perennials for cough remedy may be attributed to the rain forest nature of the location. The strong relationship between the prevailing local flora corresponds to the dominant life-form use for medicinal purpose among community members [28,51,57,58]. Ethnobotanical survey conducted in Ekiti State, which is within the same rain forest vegetation in south west of Nigeria, also indicated the dominance of woody plants for medicinal purposes among the local communities [34]. Herbaceous plant was relatively (3rd most dominant life-form) popular among the participants (Figure 3). The popularity of herb has been widely reported as a common remedy for asthma and related conditions in South Africa [28]. Furthermore, an analysis of 306 plants used for treating and managing respiratory infections and related symptoms in South Africa revealed the dominance of herbs (40.2%) which was followed by trees (35.6%) and shrubs (23.9%) as the common life-forms [50]. Likewise, Alamgeer et al. [20] indicated the dominance of herbs (57%) based on the analysis of 384 plant species used for or respiratory diseases in Pakistan. However, the seasonality that is associated with the occurrence of herbs remain a major challenge in terms of their utilisation for common ailments including cough.

As reported by the participants, diverse plant parts such as leaves, stem-bark, seeds, fruits, nuts, bulbs, and latex were utilised for managing cough in the study area (Figure 4). However, leaves (36%) were the major plant part used as a cough remedy. Factors such as ease of accessibility, availability and relative abundance of leaves often justify their dominance in traditional medicine [50,55]. Consequently, several ethnobotanical studies have observed the wide-spread utilisation of leaves for preparing herbal medicine for diverse diseases affecting humans [26,27,48,50,55,56,57]. The harvesting and utilisation of leaves or similar flora components such as fruits and seeds are often considered less destructive or detrimental to plant survival when compared to plants such as bulbs and bark [21]. From a conservation perspective, the plant species used for cough remedy in the study area are less likely to suffer from extensive population decline if continuously done in a sustainable manner.

### 3.4. Method of Preparation and Administration of Plant Species Used against Cough Associated with Respiratory Diseases

Even though the participants identified eight methods used for preparing the medicinal plants for managing cough in the study, the majority (approximately 40%) were prepared as decoction (Figure 5). In terms of the administration, all the preparations were administered orally in the study area. The popularity of decoction as a means of preparing medicinal plants for respiratory diseases was also reporting in a survey conducted in Mauritius by Suroowan and Mahomoodally [26]. Generally, decoction and infusion are often the popular means of preparing medicinal plants in local communities [20,57,64]. This may be attributed to the simplicity involved in the process whereby decoction implies the heating or boiling of the plant material in water while infusion is made by suspending the plant material in cold or pre-warmed water [26]. This described process may vary in terms of factors such as the amount of water and duration of the boiling among the different traditional healers and communities [57]. Given that these different variables influence the efficacies of the medicinal plants, the issue of adequate standardization deserves more attention from all stakeholders in traditional medicine.

Most of the plant species (for e.g., *Ananas comosus*, *Garcinia kola* and *Citrus aurantifolia*) identified were used singly, few species were in multi-herbal combinations, while in some rare cases preparations were in herbal syrup (Appendix A). In African Traditional Medicine, combining different part(s) of plants and/or plant species are common practice for treating diverse diseases [31,80,81,82].

### 3.5. Sources of Plant Species Used against Cough Associated with Respiratory Diseases

Most of the plant species used for the treatment of cough associated with respiratory diseases in the study area were collected from the wild populations (Figure 6). The heavy reliance on the wild populations of plants for medicinal purposes is well-enriched in African culture as demonstrated in several ethnobotanical survey [35,56,57]. Given that the dependence on wild population of useful plants is not sustainable over-time, the need to devise a means of mitigating this challenge has been strongly recommended [21]. For instance, adequate protection of some species can be achieved through increased enactment and implementation of appropriate laws and regulation as well as the introduction of sustainable wild harvesting methods. Traditional healers and community members are strongly encouraged to be actively involved in domestic cultivation of essential and widely-used plant species. Thus, it is commendable that some of the plants used as cough remedy are currently being cultivated in the study area (Figure 4). Similar evidence of sourcing useful plants used against different diseases has been reported by other researchers in countries such as Nigeria [56], Mauritius [26], Ghana [57], and Greece [54]. The application of biotechnological techniques also has the potential to contribute to the sustainability and commercialization of the valuable floras that are currently utilised among different ethnic groups for diverse diseases in Africa [17,21].

## 4. Conclusions

In the current study, we documented the use of 87 plant species for the management of cough associated with respiratory diseases in Ede South Local Government Area, Osun state, Nigeria. This suggests that the use of plants remain popular for treating common ailments in the study area. Similarity in the use of some plants when compared to other studies in Nigeria and other parts of the world was also established. However, a substantial (about 32%) portion of the documented plant species for management of cough indicates the rich indigenous knowledge on plant resources in the study area. The study also revealed the high dependence (63%) on the wild populations for plant materials by the local community. Given the unsustainable nature of this practice, the incorporation of cultivation and other sustainable approaches (for e.g., plant part substitution, use of leaves instead of roots/bark, where possible) to mitigate decline and eventually loss of these plants is strongly recommended. Taken together, the current findings on the medicinal uses of plant species for cough provides baseline information for future biological efficacy testing and possible isolation of biological active compounds for managing/treating cough and associated respiratory conditions as well as expectorant.

## Figures and Tables

**Figure 1 plants-09-00647-f001:**
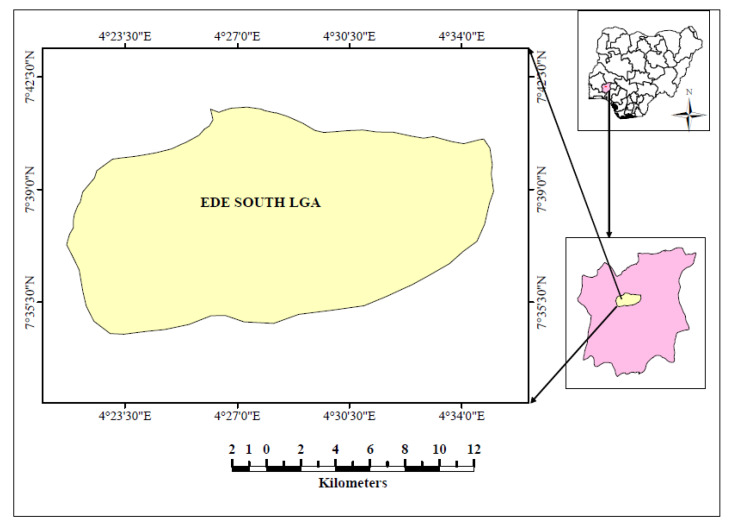
Ede South Local Government Area of Osun State, Nigeria.

**Figure 2 plants-09-00647-f002:**
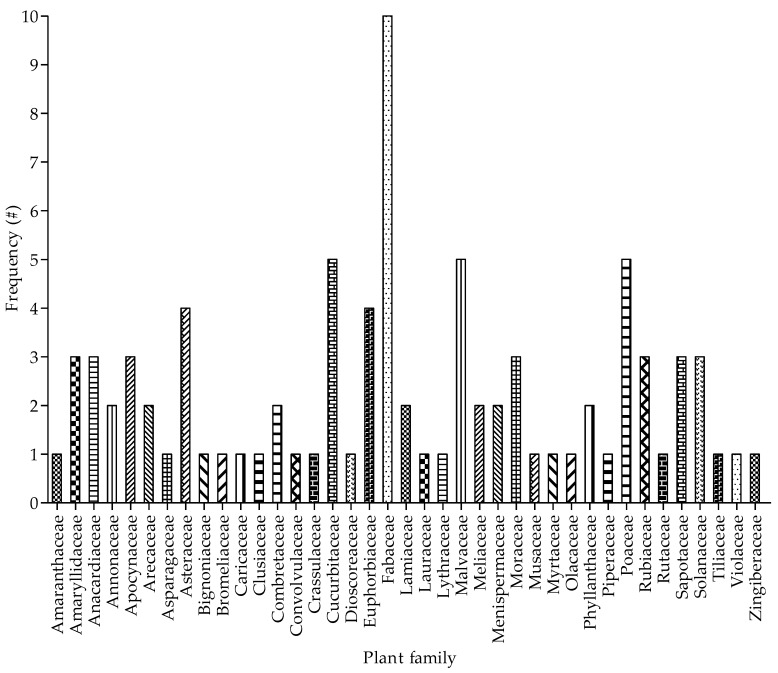
Frequency of plant family used for managing cough associated with respiratory diseases in Ede South Local Government Area of Osun State, Nigeria.

**Figure 3 plants-09-00647-f003:**
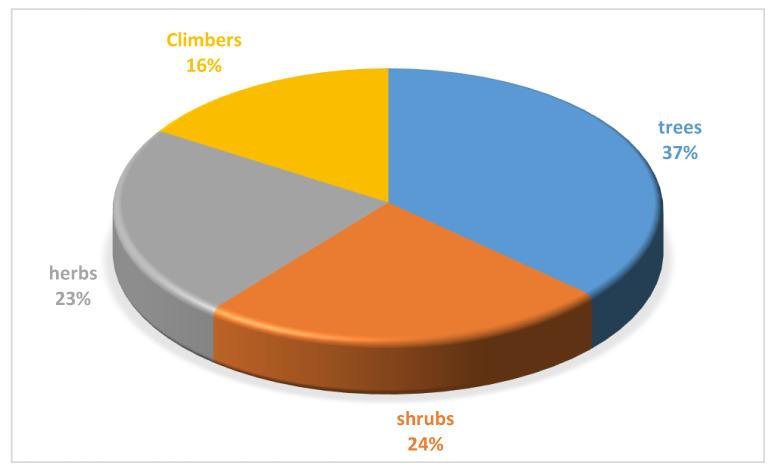
Life-form distribution (%) for the 87 plant species used for managing cough associated with respiratory diseases in Ede South Local Government Area of Osun State, Nigeria.

**Figure 4 plants-09-00647-f004:**
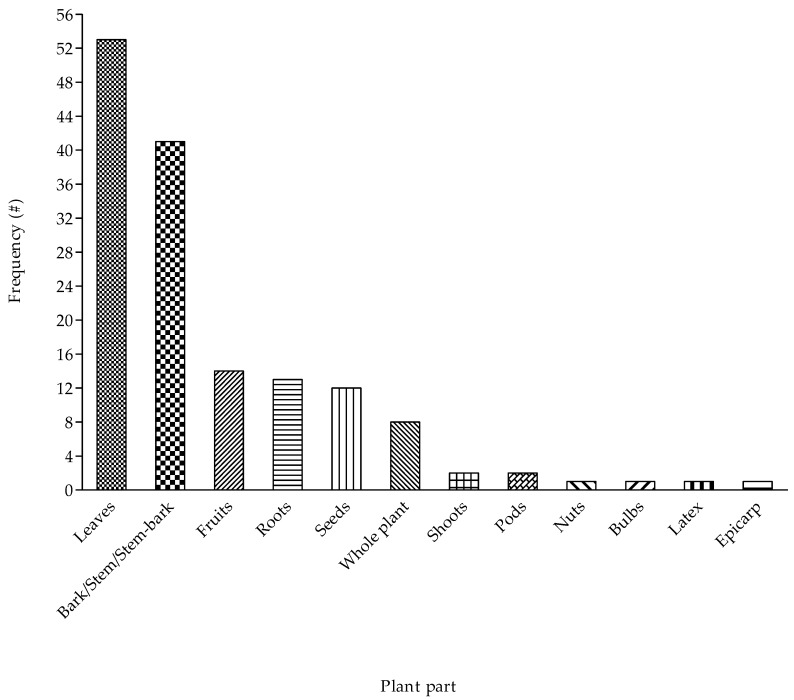
Plant parts used for managing cough associated with respiratory diseases in Ede South Local Government Area of Osun State, Nigeria.

**Figure 5 plants-09-00647-f005:**
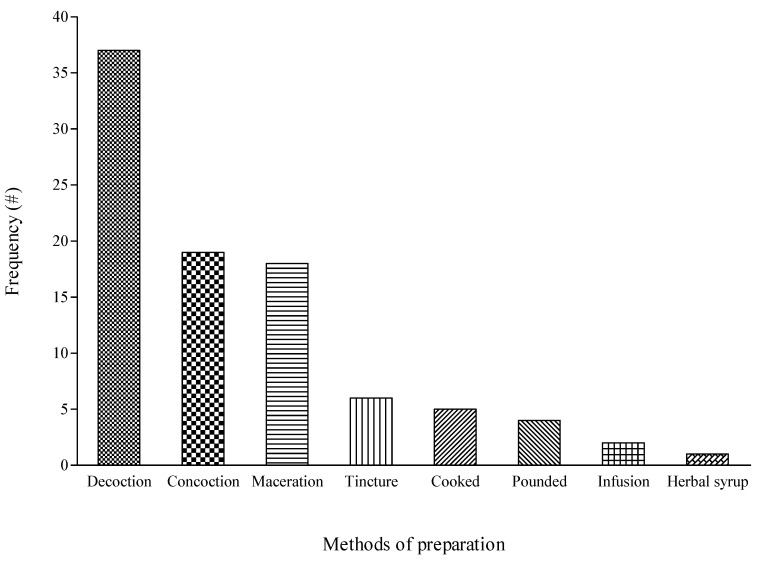
Methods of preparing plant species used for managing cough associated with respiratory diseases in Ede South Local Government Area of Osun State, Nigeria.

**Figure 6 plants-09-00647-f006:**
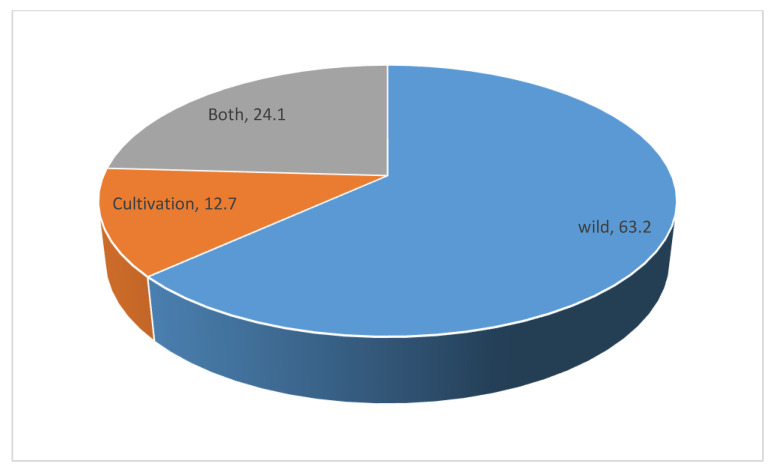
Distribution (%) of the sources of the 87 plant species used against cough associated with respiratory diseases in Ede South Local Government Area of Osun state, Nigeria.

**Table 1 plants-09-00647-t001:** Demographic characteristics of the participants (*n* = 100) in the study area.

Parameters	Participant Group	N	(%)
Gender	Male	65	65
	Female	35	35
Total		100	100
Age	<30	3	3
	31–40	15	15
	41–50	25	25
	51–60	38	38
	61–70	19	19
Total		100	100
Residence	Rural	90	90
	Urban	10	10
Total		100	100
Educational status	Primary	60	60
	Secondary	40	40
	Tertiary	0	0
Total		100	100
Occupation	Farmer	15	15
	Herbalist	35	35
	Herb seller	30	30
	Hunter	20	20
Total		100	100

**Table 2 plants-09-00647-t002:** Ethnobotanical indices (frequency of citation; FC, relative frequency of citation; RFC and fidelity level; FL) of plant species used for treating cough associated with respiratory diseases in Ede South Local Government Area of Osun State, Nigeria. (Fabaceae = Leguminosae).

Botanical Name Family, Voucher No	Local Name (Yoruba)	FC	RFC	FL (%)	Report in Nigeria	Report in Other Countries #
*Abrus precatorius* L. Fabaceae, FHI 700214	Omisinmisin, ojuologbo	10	0.10	10	Asthma [31] Bronchitis, cough, tuberculosis [29]	Asthma, cough [20] PK Cough [23] GB Respiratory diseases [24] DRC Asthma in children [25] DRC
*Aframomum melegueta* K. Schum. Zingiberaceae, FHI 610214	Ataare	10	0.10	10	Asthma [31] Cough [49] bronchitis, expectorant [29]	Tuberculosis [50] SA
*Ageratum conyzoides* (L.)L. Asteraceae, FHI 600217	Apasa/imi-esu	2	0.02	2	nrf	Cold and coughs [20] PK Respiratory diseases [24] DRC Asthma in children [25] DRC Tuberculosis [67] UG
*Allium ascalonicum* L. Amaryllidaceae, FHI 7022216	Alubosa elewe	1	0.01	1	Asthma [31] Asthma, cough [29]	nrf
*Allium sativum* L. Amaryllidaceae, FHI 714567	Ayu	1	0.01	1	Asthma [31] Asthma, cough, tuberculosis [29] Asthma, cough [47]	Respiratory tract infection [20] PK Tuberculosis [67] UG Respiratory diseases [68] MX
*Alstonia boonei* De Wild. Apocynaceae, FHI 722214	Ahun/doctor igbo	2	0.02	2	Coughs including bloody cough (tuberculosis) [36]	nrf
*Amaranthus spinosus* L. Amaranthaceae, FHI 703215	Dagunro	1	0.01	1	nrf	Bronchitis [20] PK
*Anacardium occidentale* L. Anacardiaceae, FHI 7023216	Kasu	11	0.11	11	Asthma [31,35] Tuberculosis [29]	nrf
*Ananas comosus* (L.) Merr. Bromeliaceae, FHI 722217	Ope-oyinbo	1	0.01	1	nrf	nrf
*Annickia chlorantha* (Oliv.) Setten & Maas = *Enantia chlorantha*) Annonaceae, FHI 600254	Awopa	1	0.01	1	nrf	nrf
*Antiaris toxicaria* Lesch. Moraceae, FHI 700234	Ooro	1	0.01	1	nrf	Tuberculosis [67] UG
*Asparagus africanus* Lam. Asparagaceae, FHI 700021	Aluki	1	0.01	1	nrf	Cough [48] UG
*Azadirachta indica* A. Juss. Meliaceae, FHI 605214	Dogoyaro	1	0.01	1	Tuberculosis [29,36]	Cough [20] PK Cough [64] UG Tuberculosis, cough [48,67] UG Cough [52] ID Asthma [63] PK
*Bambusa vulgaris* Schrad. Poaceae, FHI 703214	Oparun, bamboo	1	0.01	1	nrf	nrf
*Bridelia ferruginea* Benth. Phyllanthaceae, FHI 611214	Ira	10	0.10	10	Asthma [31] Whooping cough [29]	Cough [69] BE
*Bryophyllum pinnatum* (Lam.) Oken Crassulaceae, FHI 721215	Odundun	1	0.01	1	nrf	nrf
*Cajanus cajan* (L.) Millsp. Fabaceae, FHI 703216	Otili ewa	2	0.02	2	nrf	nrf
*Calotropis procera* (Aiton) Dryand Apocynaceae, FHI 705216	Bomu-Bomu	1	0.01	1	Tuberculosis and other respiratory diseases [29] Asthma, cough [35]	Asthma, cough [20,63] PK
*Capsicum annuum* L. = *Capsicum frutescens* Solanaceae, FHI 704217	Ata ijosi	1	0.01	1	nrf	nrf
*Carica papaya* L. Caricaceae, FHI 710213	Ibepe	1	0.01	1	nrf	Cough, tuberculosis [48,64,70] UG Respiratory diseases [24] DRC Asthma in children [25] DRC Cough [71] SA
*Chasmanthera dependens* Hochst. Menispermaceae, FHI 711215	Agba-ato, Ato	1	0.01	1	Asthma [31]	Children’s cough [72] UG
*Chromolaena odorata* (L.) R.M.King & H.Rob. Asteraceae, FHI 705209	Akintola	1	0.01	1	nrf	nrf
*Chrysophyllum albidum* G.Don Sapotaceae, FHI 730274	Agbalumo	2	0.02	2	Asthma [31]	nrf
*Cissampelos owariensis* P.Beauv. ex DC. Menispermaceae, FHI 723217	Jokojee, Jenjokoo	1	0.01	1	Lung diseases [47]	nrf
*Citrullus colocynthis* (L.) Schrad. Cucurbitaceae, FHI 722216	Egusi’baara	1	0.01	1	nrf	Bronchial asthma [20] PK
*Citrus aurantiifolia* (Christm.) Swingle Rutaceae, FHI 700316	Oronbo wewe/ osan wewe	10	0.10	10	nrf	Cough [64] UG
*Cocos nucifera* L. Arecaceae, FHI 711215	Agbon	2	0.02	2	Asthma [31] Bronchitis [35,47]	nrf
*Cola acuminata* (P.Beauv) Schott & Endl. Malvaceae, FHI 730214	Obi gidi	1	0.01	1	nrf	nrf
*Combretum bracteatum* (C.Lawson) Engl. & Diels Combretaceae, FHI 705215	Ogan dudu	1	0.01	1	nrf	nrf
*Combretum micranthum* G. Don Combretaceae, FHI 723214	Ogan	1	0.01	1	nrf	nrf
*Corchorus olitorius* L. Malvaceae, FHI 730219	Ewedu	1	0.01	1	nrf	nrf
*Crinum jagus* (J.Thomps.) Dandy Amaryllidaceae, FHI 705215	Ogede odo	12	0.12	12	Asthma [31,73] Cough [30] Tuberculosis [47,73]	nrf
*Crotalaria retusa* L. Fabaceae, FHI 705216	Koropo	1	0.01	1	nrf	nrf
*Crudia klainei* De Wild. Fabaceae, FHI 710121	Afomo	2	0.02	2	nrf	nrf
*Cymbopogon citratus* (DC.) Stapf, Poaceae, FHI 605214	Ewe tea	3	0.03	3	nrf	Cough, tuberculosis [67,70] UG Respiratory diseases [68] MX Respiratory infections [74] UG Cough, bronchitis, asthma, respiratory diseases [26] MU
*Dioclea reflexa* Hook.f. Fabaceae, FHI 705213	Agbaarin	2	0.02	2	Asthma [47]	Respiratory diseases [24] DRC Respiratory diseases in children [25] DRC
*Dioscorea dumetorum* (Kunth) Pax Dioscoreaceae, FHI 765217	Esuru	1	0.01	1	nrf	Respiratory diseases [24] DRC Respiratory diseases in children [25] DRC
*Elaeis guineensis* Jacq. Arecaceae, FHI 600215	Ope	1	0.01	1	Asthma, bronchitis, chest pain, tuberculosis [29]	Respiratory diseases [24] DRC Wet cough in children [25] DRC
*Erythrophleum suaveolens* (Guill. & Perr.) Brenan Fabaceae, FHI 744214	Igi obo, Eru-obo	1	0.01	1	nrf	nrf
*Ficus exasperata* Vahl Moraceae, FHI 611002	Eepin	2	0.02	2	Coughs including bloody cough (tuberculosis) [36]	Respiratory diseases [75] TA
*Ficus platyphylla* Delile Moraceae, FHI 724234	Obobo	1	0.01	1	Coughs including bloody cough (tuberculosis) [36]	nrf
*Garcinia kola* Heckel Clusiaceae, FHI 705215	Orogbo	10	0.10	10	Asthma [31,35] Cough [49] Cough, tuberculosis [29] Coughs including bloody cough (tuberculosis) [36]	nrf
*Gossypium barbadense* L. Malvaceae, FHI 720221	Owu	2	0.02	2	Asthma [31]	nrf
*Hybanthus enneaspermus* (L.) F. Muell. Violaceae, FHI 704216	Abiwere	1	0.01	1		nrf
*Ipomoea batatas* (L.) Lam. Convolvulaceae, FHI 711025	Anomo, Odunkun	2	0.02	2	Asthma [35]	nrf
*Jatropha curcas* L. Euphorbiaceae, FHI 740216	Lapalapa funfun	10	0.10	10	Tuberculosis [29]	Bronchitis [20] PK Respiratory diseases [24] DRC Respiratory diseases in children [25] DRC
*Jatropha gossypiifolia* L. Euphorbiaceae, FHI 600215	Lapalapa pupa	1	0.01	1	nrf	nrf
*Jatropha multifida* L. Euphorbiaceae, FHI 603214	Ogege	10	0.10	10	nrf	nrf
*Khaya grandifoliola* C.DC. Meliaceae, FHI 770215	Oganwo	11	0.11	95.0	Coughs including bloody cough (tuberculosis) [36]	nrf
*Kigelia africana* (Lam.) Benth. Bignoniaceae, FHI 702004	Pandoro	1	0.01	11	Asthma [31] Cough [47]	Cough [48,64] UG Tuberculosis [67] UG Tuberculosis and other respiratory diseases [76] CM
*Lagenaria breviflora* (Benth.) Roberty Cucurbitaceae, FHI 710218	Tagiri	2	0.02	2	nrf	nrf
*Launaea taraxacifolia* (Willd.) Amin ex C.Jeffrey Asteraceae, FHI 730211	Yanrin	1	0.01	1	nrf	nrf
*Lawsonia inermis* L. Lythraceae, FHI 702616	Laali	2	0.02	2	Asthma, tuberculosis [29]	Cough, bronchitis [20] PK
*Luffa cylindrica* (L.) M.Roem. Cucurbitaceae, FHI 710216	Kankan-ayaba	1	0.01	1	Hemoptysis [29]	nrf
*Mangifera indica* L. Anacardiaceae, FHI 700209	Mongoro	2	0.02	2	nrf	Asthma, cough [20] PK Cough [77] BR Cough [64] UG Respiratory diseases [68] MX
*Mimosa pudica* L. Fabaceae, FHI 710217	Patanmo	1	0.01	1	nrf	Asthma [20] PK
*Momordica charantia* L. Cucurbitaceae, FHI 722217	Ejinrin	1	0.01	1	nrf	Tuberculosis [67] UG
*Morinda lucida* Benth. Rubiaceae, FHI 700514	Oruwo	1	0.01	1	nrf	Bronchitis, cough [20] PK
*Mucuna pruriens* (L.) DC. Fabaceae, FHI 733214	Werepe	1	0.01	1	nrf	Tuberculosis and other respiratory diseases [76] CM
*Musa paradisiaca* L. Musaceae, FHI 733215	Ogede agbagba	1	0.01	1	nrf	Whooping cough [20] PK Cough [64] UG
*Nicotiana tabacum* L. Solanaceae, FHI 711211	Taba	1	0.01	1	Asthma [31] Bronchitis [29]	Asthma [78] SP
*Ocimum americanum* L. (*Ocimum canum*) Lamiaceae, FHI 734512	Efinrin wewe	1	0.01	1	Cough, chest pain [29] Cough [35,47] Coughs including bloody cough (tuberculosis) [36]	Cough [79] UG Respiratory diseases [24] DRC Respiratory diseases in children [25] DRC
*Ocimum gratissimum* L. Lamiaceae, FHI 7009001	Efinrin nla	1	0.01	1	nrf	Respiratory diseases [24] DRC Cough [51] ET
*Parkia biglobosa* (Jacq.) G.Don Fabaceae, FHI 709123	Iru, igba	2	0.02	2	Coughs including bloody cough (tuberculosis) [36]	
*Persea americana* Mill. Lauraceae, FHI 705215	Igi pia	2	0.02	2	nrf	Cough, respiratory infectious [48,79] UG Tuberculosis [67] UG
*Phyllanthus amarus* Schumach. & Thonn. Phyllanthaceae, FHI 715214	Eyin-olobe	1	0.01	1	nrf	Tuberculosis [67] UG
*Picralima nitida* (Stapf) T. Durand & H.Durand Apocynaceae, FHI 721218	Abere	1	0.01	1	Asthma [31]	nrf
*Piper guineense* Schumach. & Thonn. Piperaceae, FHI 703217	Iyere	1	0.01	1	Tuberculosis, whooping cough [29] Coughs including bloody cough (tuberculosis) [36]	nrf
*Plukenetia conophora* Müll.Arg. Euphorbiaceae, FHI 712343	Awusa, asala	1	0.01	1	nrf	nrf
*Psidium guajava* L. Myrtaceae, FHI 712215	Gilofa	1	0.01	1	Tuberculosis [29] Coughs including bloody cough (tuberculosis) [36] Cough [35]	Old cough, bronchitis, chronic whooping cough [63] PK Cough [64,65] UG Cough [27] SA Cough [66] ZB
*Saccharum officinarum* L. Poaceae, FHI 742217	Ireke	1	0.01	1	Asthma [31]	Respiratory diseases [24] DRC Respiratory diseases in children [25] DRC
*Sarcocephalus latifolius* (Sm.) E.A.Bruce Rubiaceae, FHI 720216	Egbesi	1	0.01	1	Cough [29]	nrf
*Solanum americanum* Mill. Solanaceae, FHI 702200	Efo odu	1	0.01	1	nrf	nrf
*Sorghum bicolor* (L.) Moench Poaceae, FHI 734017	Oka baba	1	0.01	1	nrf	nrf
*Spermacoce verticillata* L. = *Borreria verticillata* (L.) G.Mey Rubiaceae, FHI 710210	Irawo’le	1	0.01	1	nrf	nrf
*Spondias mombin* L. Anacardiaceae, FHI 705215	Iyeye	1	0.01	1	Asthma, tuberculosis [29]	nrf
*Sterculia rhinopetala* K.Schum. Malvaceae, FHI 720216	Itako	1	0.01	1	nrf	nrf
*Strombosia grandifolia* Hook.f. ex Benth. Olacaceae, FHI 710017	Itako pupa	1	0.01	1	nrf	nrf
*Synsepalum dulcificum* (Schumach. & Thonn.) Daniell Sapotaceae, FHI 700210	Agbayun	1	0.01	1	nrf	nrf
*Talinum fruticosum* (L.) Juss. (Syn: *Talinum triangulare*) Talinaceae, FHI 700347	Gbure	1	0.01	1	nrf	nrf
*Telfairia occidentalis* Hook.f. Cucurbitaceae, FHI 700234	Eweroko, Ugu	1	0.01	1	nrf	nrf
*Tetrapleura tetraptera* (Schum. & Thonn.) Taub. Fabaceae, FHI 701354	Aridan	2	0.02	2	Asthma [31] Coughs including bloody cough (tuberculosis) [36]	nrf
*Theobroma cacao* L. Malvaceae, FHI 700314	Koko	1	0.01	1	nrf	nrf
*Uvaria afzelii* G.F. Scott-Elliot Annonaceae, FHI 700334	Gbogbonise	1	0.01	1	For any ailment [73]	nrf
*Vernonia amygdalina* Delile Asteraceae, FHI 700167	Ewuro	2	0.02	2	Asthma, cough, tuberculosis [29]	Tuberculosis [67] UG Cough [48] UG Respiratory infections [74] UG
*Vitellaria paradoxa* C.F.Gaertn. Sapotaceae, FHI 700218	Emiyemi, Ori	1	0.01	1	Asthma, expectorant, tuberculosis, whooping cough [29]	nrf
*Zea mays* L. Poaceae, FHI 700228	Agbado	1	0.01	1	Cough [29]	Cough [20] PK Asthma [28] SA

# Other Countries: BE = Benin, BR = Brazil, ET = Ethiopia, CM = Cameroun, DRC = Democratic Republic of Congo, GB = Gabon, ID = India, MU = Mauritius, MX = Mexico, SA = South Africa, SP= Spain, PK = Pakistan, TA = Tanzania, UG =Uganda, ZB = Zimbabwe, nrf = no report found.

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
