# Peer review of "Ethnobotanical Survey of Plants Used for Treating Cough Associated with Respiratory Conditions in Ede South Local Government Area of Osun State, Nigeria"

_plants, 2020, doi:10.3390/plants9050647_

Round 1

Reviewer 1 Report

Land use map of the study site presents more info regarding built up areas, settlement, forest, agriculture and water body. 

Reviewer 2 Report

This manuscript, contributes interesting data on plants used against cough in a region of Nigeria, is a revised version of one I evaluated in its original version. I find the article in general, interesting, and the authors have considerably improved the manuscript in the revision. I advise now a minor revision, and I provide the authors with some comments in order to help them to prepare an improved version.

1.- The authors used, in the first version, a high number of quantitative indices, what I found interesting. Now they have dramatically reduced this part, probably as per the advice of another referee, what is understandable. In any case (and now indeed, because of this diminution of indices in the manuscript) I insist advising them to include also the informant consensus factor (R. T. Trotter and M. H. Logan, “Informant consensus: a new approach for identifying potentially effective medicinal plants,” in Plants in Indigenous Medicine and Diet, Behavioural Approaches, N. L. Etkin, Ed., pp. 91–112, Redgrave Publishing Company, Bredford Hills, New York, 1986), which is a good indicator of the robustness of the work performed and the dataset obtained. The authors state in their response to my comments to the first version, that this index cannot be calculated for a particular plant use. I believe it can, and it has been done, for instance in a paper with a similar focus (respiratory system ailments) than the present one: Rigat, M., Vallès, J., Iglésias, J., Garnatje, T. 2013. Traditional and alternative natural therapeutic products used in the treatment of respiratory tract infectious diseases in the eastern Catalan Pyrenees (Iberian Peninsula). Journal of Ethnopharmacology, 148: 411-422.

2.- The authors state that “Most of the 87 plants documented (…) were from family Fabaceae”, but I counted 10 plants belonging to this family, and this is not most of 87. They could rather affirm that the family most represented in this survey is Fabaceae, but not that most plants recorded belong to it.

3.- I believe that several types of information now only presented in the supplementary table are the fruit of the ethnobotanical prospection and should appear in table 1: local name, part used and method of preparation. The remaining parts are adequate fporor the supplementary table.

4.- Formal aspects. Line 22: write “Information obtained was analyzed” instead of “Information obtained was analyzed”. Line 217: write “have been reported” instead of “have been report”.

Reviewer 3 Report

Upon reviewing this manuscript again I have been pleasantly surprised by the work done by the authors.
All the comments I made in each section of the manuscript have been taken into account. Specific:

Introduction: It has been greatly improved. I consider that ethnobotanical information on Africa, Nigeria and the study area should be included. This would greatly increase the interest of the manuscript.

Methodology: This is one of the sections that has been modified the most. Indices that did not contribute anything to the study have been eliminated and other very good ones have been included, such as that of Tardio et al. 2008.

Results and Discussion: They have a much higher value thanks to the indexes used in the analysis. The results obtained when compared with other ethnobotanical backgrounds in Nigeria and Africa should be highlighted.

Conclusion: and much more accurate now, where the novel results are highlighted from the ethnobotanical point of view.

I believe that the authors should make an effort to collect bibliographic information on traditional knowledge existing in Nigeria. These studies would revalue the Flora of the country and future studies will have a higher quality.

Author Response

This manuscript is a resubmission of an earlier submission. The following is a list of the peer review reports and author responses from that submission.

Round 1

Reviewer 1 Report

Dear authors

In spite of being of great interest, your paper is poorly written and methods (ethnobotanical indices) should be justified or simplified.

You find my comments within the file here attached.

Reviewer 2 Report

I thank editors, MDPI Plants journal for providing this opportunity to review this article entitled “Ethnobotanical survey of plants used as cough remedy in Ede South Local Government Area of Osun State, Nigeria”. The paper is novel and interdisciplinary and possesses of global significance.

However, the paper was found hastily written because the results were under-discussed. It is worth considering applying the following points while revising and resubmitting before publishing in the journal Plants.

The major findings of ecological/diversity indices and ethnobotanical indices (use mentions, Ranking, Fidelity Level) were under-described and their association was little discussed. Besides this issue, some minor comments are described below.

Abstract: Line 23: assessed

Line 28: Its not 87 plants, its 87 species

Introduction:

Why such type of study is important in Nigeria and how and why problematic the cough is there? Please describe them in your introduction to contextualize your study.

Study area:

Line 95: According to the government document, what are the common diseases of Ede South LGA of Osun State Nigeria?

Methods:

Line 113: What is the sampling intensity and strategy? What is the composition of 100 respondents by age group, gender, occupation, etc? Or simply saying, how many were male? How many were TM practitioners? How many were above age 70?

Line 121: I suggest authors to use latest theplantlist.org

Result and Discussion

Line: 195-196, should go to the Method section.

I suggest relocating Table 1 in method section.

Table 2. FL

What does the Pr stand for? There is PR in abstract section. Please use abbreviations in standard and consistent way.

Table 3.

Should it be Density? If its Frequency, how did you compute Frequency?

There are a good number of studies assessing the relationship of ecological/diversity indices and ethnobotanical indices. However, in the present study, how the ethnobotany is associated with diversity/ecological indices is under-discussed.

One study carried out by Bussmann et al 2020 https://doi.org/10.1016/j.ecolind.2019.105679

describing ethnobotanical-ecological association is pertinent to refer in the light of your study findings. I encourage you to refer and follow his way of discussion.

Why families, Amarillydaceae, Anacardiaceae and Meliaceae were found significant for treatment of cough? Do these families and their species possess expectorant activities?

Reviewer 3 Report

This manuscript contributes interesting data on plants used against cough in a region of Nigeria. The work is well designed and performed, the methods seem to have been adequately used, the results are good in number and quality and they are discussed on the frame of a pertinent literature set. The work falls within the scope of the journal and, in my opinion, it is worthy of publication, but not its present form, but after a revision that I qualify of major, but that I find easily feasible given the general quality of the work. I provide now the authors with some comments in order to help them to prepare an improved version.

1.- The field interview methods should be slightly more explained. The authors allude to a questionnaire, but they do not explain what was the interview type: semistructured, closed questionnaire…. In case this is a closed questionnaire, I believe it would be good presenting it as a supplementary material.

2.- Still in the methods, the language in which the interviews were developed should be indicated, as well as the need or not for interpreters.

3.- The authors use a high number of quantitative indices, what is interesting. I advise them to include also the informant consensus factor (R. T. Trotter and M. H. Logan, “Informant consensus: a new approach for identifying potentially effective medicinal plants,” in Plants in Indigenous Medicine and Diet, Behavioural Approaches, N. L. Etkin, Ed., pp. 91–112, Redgrave Publishing Company, Bredford Hills, New York, 1986), which is a good indicator of the robustness of the work performed and the dataset obtained.

4.- In table 2, the language to which the local names belong should be indicated.

5.- I do not see any sense in indicating, in table 2, the so-called common names, being in fact the English names, unless they have been collected by the authors during the ethnobotanical interviews. If this is not the case, English names are not necessary, the unequivocality being ensured by the Latin scientific names.

6.- The authors write they follow The Plant List for taxonomy and nomenclature. They should carefully revise all names in order to avoid cases such as “Hook f.” or “J.C. Wendel” (in The Plant List): “Hook.f.”, “J.C.Wendel”) or “Synsepalum dulcificum (Schumach. & Thonn.) William Freeman Danielferl” (in The Plant List: “Synsepalum dulcificum (Schumach. & Thonn.) Daniell”). Additionally, revise all scientific names in order they are italicised and the authorities are neither italicised nor underlined (this happens in some cases).

7.- In Table 2, a column with the indication of the number of use-reports (or frequency of utilisation) of each taxon would be informative.

8.- Indicate the full name of all acronyms (LGA, PR and others) the first time they appear in the abstract and in the rest of the text.

9.- I believe the results and discussion section would benefit, to be easier to follow, of some more subheadings, for instance parts of plants used and types of preparation forms, which are now included in the very generic plants used for cough.

10.- Some information on previous ethnobotanical prospections in the studied area or the surroundings would be informative and, if any, the results could be compared to those obtained now by the authors.

11.- Does the distribution of the recorded plants in plant growth forms fit with the general abundance of each kind d of grow form in the territory?

12.- A reference is needed in the sentence starting “In addition, the  recognition of herbs”.

13.- It is not clear for me whether the diversity indices are applied to the 87 plants recorded in the prospection or to all the flora of the area studied. In this manuscript, it would be logical the first case, but sentences such as “This implies that there was higher species dominance and low species richness in the study area” and “The diversity indices obtained in this study showed a lesser diverse ecosystem and the relative density of the species was low” make me think on the other one. I believe the link of diversity indexes applied to the whole flora of the area studied with the data obtained in the prospection should appear clearly. In addition, the number of taxa of the total flora and there references from which this information comes should be indicated.

14.- Formal aspects. Do not write common names (taxa, tussis, ward, concentration, diversity…) with capital initials unless they start a sentence. Write taxa, Margalef, Asparagaceae, Asteraceae instead of taxas, Margelef, Asperagaceae, Astereceae.

Reviewer 4 Report

At first, the manuscript entitled " Ethnobotanical survey of plants used as cough  remedy in Ede South Local Government Area of Osun  State, Nigeria" identified a work of relevant ethnobotanical interest. However, by reading the text carefully he found serious methodological problems that were difficult to solve.

Here are the comments to the manuscript:

Title: it would have been better to talk about the plants used for respiratory system problems than cough remedies.

Introduction:

- It says nothing about the vegetation of the study area.

- 42-79: very extensive and little necessary. It would have to be summed up a lot.

- 80-92: It is relatively good, but ethnobotanical information is missing from Africa, Nigeria and the study area. It is essential to know what has been done to raise the study and know if our results are novel.

- 91-92: the objective is not very concrete.

Material and method:

Study area: What vegetation is there? Do you know the Flora of this area? It is essential to answer these questions.

112-117: The group of people interviewed is very interesting.

121-123: very well that there are herbarium witnesses of all plants.

124-128: Good.

129-192: Analysis of ethnobotanical data:

The methodology used is not adequate. Use ecological indexes that are useless in this type of work. I should have applied quantitative ethnobotany indices such as those collected by Silva et al. (2006) Only Alesiades (1996) is suitable.

* SILVA, V. A. D., L. H. C. ANDRADE & U. P. ALBURQUERQUE. 2006. Revising the cultural significance index: the case of the fulni-ô in northeastern Brazil. Field Methods 18 (1): 96-108.

Results and Discussion:

194-203: Include it in Material and methods

204-228: Merely descriptive. The interesting thing would be to know if the results obtained are novel for the region, Nigeria, Africa or the World.

229: Table 2 would include it in an Appendix.

230-287: Merely descriptive. It contributes little. The interesting thing would be to know if the results obtained are novel for the region, Nigeria, Africa or the World.

291-368: the discussion on ecological indices does not contribute anything to a work on Ethnobotany.

Conclusion:

It is not interesting. In addition, he says that the collections of these medicinal plants are bad for the ecosystem. But this does not quantify it. Wild plant collectors, if they have traditional knowledge, do not usually cause damage to habitats, as it is their livelihood. The dangerous thing is the destruction of the habitat.

The interest of the results of this manuscript could be great, but the methodology used has ruined the work. Therefore, I consider that the work must be rejected. Despite this, I encourage the authors to rewrite the work again, but for this they need a thorough knowledge of current ethnobotanical methodologies.